# M13 phage grafted with peptide motifs as a tool to detect amyloid-β oligomers in brain tissue

Ivone M. Martins [1,2,3,4,8✉], Alexandre Lima[1,2,4,8], Wim de Graaff[4], Joana S. Cristóvão[5,6], Niek Brosens[4], Eleonora Aronica [7], Leon D. Kluskens[1,2,9], Cláudio M. Gomes [5,6], Joana Azeredo [1,2] & Helmut W. Kessels [3,4✉]

Oligomeric clusters of amyloid-β (Aβ) are one of the major biomarkers for Alzheimer's disease (AD). However, proficient methods to detect Aβ-oligomers in brain tissue are lacking. Here we show that synthetic M13 bacteriophages displaying Aβ-derived peptides on their surface preferentially interact with Aβ-oligomers. When exposed to brain tissue isolated from APP/PS1-transgenic mice, these bacteriophages detect small-sized Aβ-aggregates in hippocampus at an early age, prior to the occurrence of Aβ-plaques. Similarly, the bacteriophages reveal the presence of such small Aβ-aggregates in *post-mortem* hippocampus tissue of AD-patients. These results advocate bacteriophages displaying Aβ-peptides as a convenient and low-cost tool to identify Aβ-oligomers in *post-mortem* brain tissue of AD-model mice and AD-patients.

[1] CEB- Centre of Biological Engineering, University of Minho, 4710-057 Braga, Portugal. [2] LABBELS – Associate Laboratory, Braga/Guimarães, Portugal. [3] Netherlands Institute for Neuroscience, Amsterdam, the Netherlands. [4] Swammerdam Institute for Life Sciences, University of Amsterdam, Amsterdam Neuroscience, Amsterdam, the Netherlands. [5] Biosystems & Integrative Sciences Institute, Faculdade de Ciências, Universidade de Lisboa, Lisboa, Portugal. [6] Departamento de Química e Bioquímica, Faculdade de Ciências, Universidade de Lisboa, Lisboa, Portugal. [7] Amsterdam UMC location University of Amsterdam, Department of (Neuro)Pathology, Amsterdam Neuroscience, Amsterdam, the Netherlands. [8]These authors contributed equally: Ivone M. Martins, Alexandre Lima. [9]Deceased: Leon D. Kluskens. ✉email: ivone.martins@ceb.uminho.pt; h.w.h.g.kessels@uva.nl

Alzheimer's disease (AD) is a chronic, progressive, degenerative brain disorder and the most common type of dementia. The prime suspect to cause AD is the small-sized amyloid-beta (Aβ) peptide, released by neurons after β and γ secretase cleavage of the amyloid precursor protein (APP), a transmembrane protein abundantly present in the central nervous system. APP cleavage results in a heterogeneous group of peptides of varying length, with the 40 and 42 amino acid-long isoforms as the two main toxic species. Aβ42 showed to be more prone to aggregation when compared with Aβ40, and a higher presence of this residue has been correlated with higher neurotoxicity[1]. In pathological conditions, Aβ peptides progressively aggregate into soluble oligomers and protofibrils, and deposit as insoluble amyloid fibrils into plaques[2], which are a hallmark of AD. Nevertheless, it has become increasingly clear that it is not Aβ immobilized in plaques but instead the still-soluble oligomeric forms that are the toxic species of Aβ[3,4]. Before amyloid plaques can be detected in the brains of APP-transgenic mice, Aβ-oligomers can trigger the loss of synapses[1]. Synapse loss correlates strongly with cognitive decline during early AD[5]. It is therefore relevant to correlate the level of synapse loss and cognitive decline with the presence of Aβ-oligomers instead of amyloid plaques in the brain. Unfortunately, whereas immunohistochemical tools that detect Aβ-plaques are available, we are currently destitute of tools that selectively detect Aβ-oligomers in brain tissue.

Peptides hold a great potential to selectively target oligomeric Aβ[6]. Elegant previous studies showed that Aβ-peptide motifs corresponding to amino acids 30-39 ($^{30}$AIIGLMVGGV$^{39}$) or 33-42 ($^{33}$GLMVGGVVIA$^{42}$) interact with oligomers and fibrils, but not monomers, with nanomolar affinity[7]. However, for use in immunohistochemistry, peptides need to be immunochemically labeled while retaining the architectural structure that confers target recognition and specificity. When grafted into the complementarity-determining regions of antibodies, these peptides were shown to neutralize Aβ toxicity[7]. We here tested whether these Aβ-specific peptides can detect Aβ-oligomers and fibrils when displayed on bacteriophages (phages).

Phages are viruses of bacteria and are therefore safe for humans, easy and cheap to produce at large scale, and used as an antibiotic in clinical practice since the 1920s. Phages are also an attractive biotechnological tool for biomedical research because they can be genetically and/or chemically modified to display a variety of biomolecules on their surface[8,9]. This report describes the capacity of synthetic phages that display Aβ-specific peptides on their surface to target Aβ-aggregates in vitro and in immunohistochemistry of mouse and human *post-mortem* brain tissue.

## Results and discussion

**Engineered bacteriophages preferentially recognize Aβ-oligomers.** We engineered M13 filamentous phage to display Aβ30-39 ($^{30}$AIIGLMVGGV$^{39}$) or Aβ33-42 ($^{33}$GLMVGGVVIA$^{42}$) peptides onto coat protein III. These synthetic phage particles, which we named AB30-39 and AB33-42 respectively, display 5–8 copies of peptide motifs on their surface clustered at the end of the filamentous phage (Fig. 1a). To test whether they interact with Aβ-aggregates, we monitored their effect on Aβ-aggregation kinetics using thioflavin T (ThT), an amyloid-sensitive fluorescent dye whose intensity correlates with fibril mass. Aβ-fibril formation proceeds via complex kinetics through which Aβ42 monomers self-assemble into oligomers and fibrils via a secondary nucleation mechanism[10] (Fig. 1b), resulting in sigmoid-shaped curve consisting of a lag-phase, fibril-elongation phase, and plateau phase (Fig. 1c)[11]. We tested and compared the effects of the engineered phages AB30-39 and AB33-42 at increasing

titers ($1 \times 10^8$, $1 \times 10^9$, and $1 \times 10^{10}$ plaque-forming units (pfu) ml$^{-1}$) on Aβ42 aggregation *versus* those of the empty M13 bacteriophage control (Fig. 1d, e).

At high titers of $1 \times 10^9$ and $1 \times 10^{10}$ pfu ml$^{-1}$, control M13-phages lowered the plateau phase of fibril formation (Fig. 1d, e), indicating they affected aggregation kinetics, which relates to a previous finding that M13-phage by itself binds Aβ-aggregates[12]. AB30-39 and AB33-42 further influenced the formation of fibrils compared with control M13, although each differently (Fig. 1d, e). AB30-39 results in a slight increase in the reaction half-time and in a decrease in the amount of formed fibrils as denoted by a decrease in the end-point ThT intensity, which translates to a lower fibril mass. On the other hand, in the presence of AB33-42 phages, the reaction half-time remains identical to that of the M13 control while fibril mass at the end-point of the reaction at the higher phage titer is comparable to that of M13 at higher phage concentrations. These distinct effects of AB30-39 versus AB33-42 likely have a rationale in the context of the structural characteristics of Aβ42 amyloid fibrils[13,14]: it has been pointed out that the hydrophobic strip formed by residues Val40 and Ala42 that run down the outer surface of the protofilament enhance secondary nucleation. This leads to the speculation that, since AB33-42 contains these two residues, it is binding to Aβ42-oligomers might in theory contribute to fibril formation, whereas AB30-39 binding may interfere with aggregation by lacking these c-terminal amino acids, slightly increasing the reaction half time and decreasing fibril mass (Fig. 1e).

At a relatively lower concentration of $1 \times 10^8$ pfu ml$^{-1}$, empty M13 did not affect fibril aggregation, whereas AB30-39 and AB33-42 reduced plateau levels without affecting lag-phase or reaction half-time of fibril formation as measured by ThT (Fig. 1d, e). These results suggest that, under these conditions, they predominantly sequester Aβ-oligomers from participating in fibril formation[15]. AB33-42 lowered end-point fibril mass further than AB30-39 did, which agrees with AB33-42 having a higher affinity for Aβ-oligomers than AB30-39[7]. These experiments indicate that, at a concentration of $1 \times 10^8$ pfu ml$^{-1}$, AB30-39 and AB33-42 preferentially recognize Aβ-oligomers.

**Engineered bacteriophages detect Aβ-oligomers in brain tissue from APP/PS1-transgenic mice.** We next examined whether the engineered phages are capable of detecting Aβ-aggregates produced in brain tissue from APP/PS1-transgenic mice. We isolated the brains of young adults (3-4-month-old) and aged (10–12-month-old) APP/PS1-transgenic and wild-type mice and performed immunohistochemistry on brain slices. Although APP/PS1-mice show synaptic deficits and memory impairment when 3–4-month-old[16–18], amyloid-plaques were detected by anti-Aβ monoclonal antibodies (Aβ-mAbs) in the hippocampus of aged APP/PS1-mice but of not young adults (Fig. 2a). Brain slices were exposed to $1 \times 10^8$ pfu ml$^{-1}$ of phages and subsequently stained with anti-bacteriophage mAbs. Whereas M13 controls did not show staining, both AB30-39 and AB33-42 phages revealed punctate staining in hippocampus of APP/PS1-mice, and the density of these puncta increased with age (Fig. 2a,b). These puncta were ≤1 μm in size, suggesting predominantly oligomers up to the size of protofibrils were detected, but not larger fibrils or amyloid-plaques. In line with Aβ33-42 peptides having a higher affinity for Aβ-oligomers than Aβ30-39[7], AB33-42 consistently detected more puncta compared with AB30-39 (Fig. 2b). Interestingly, these puncta were predominantly present around cell bodies in *stratum pyramidale*, but not in dendritic areas in *stratum radiatum* in slices from young adults (Fig. 2a, c). In slices from aged mice, puncta were evenly distributed among cell body and dendritic areas (Fig. 2a, c). Possibly the production,

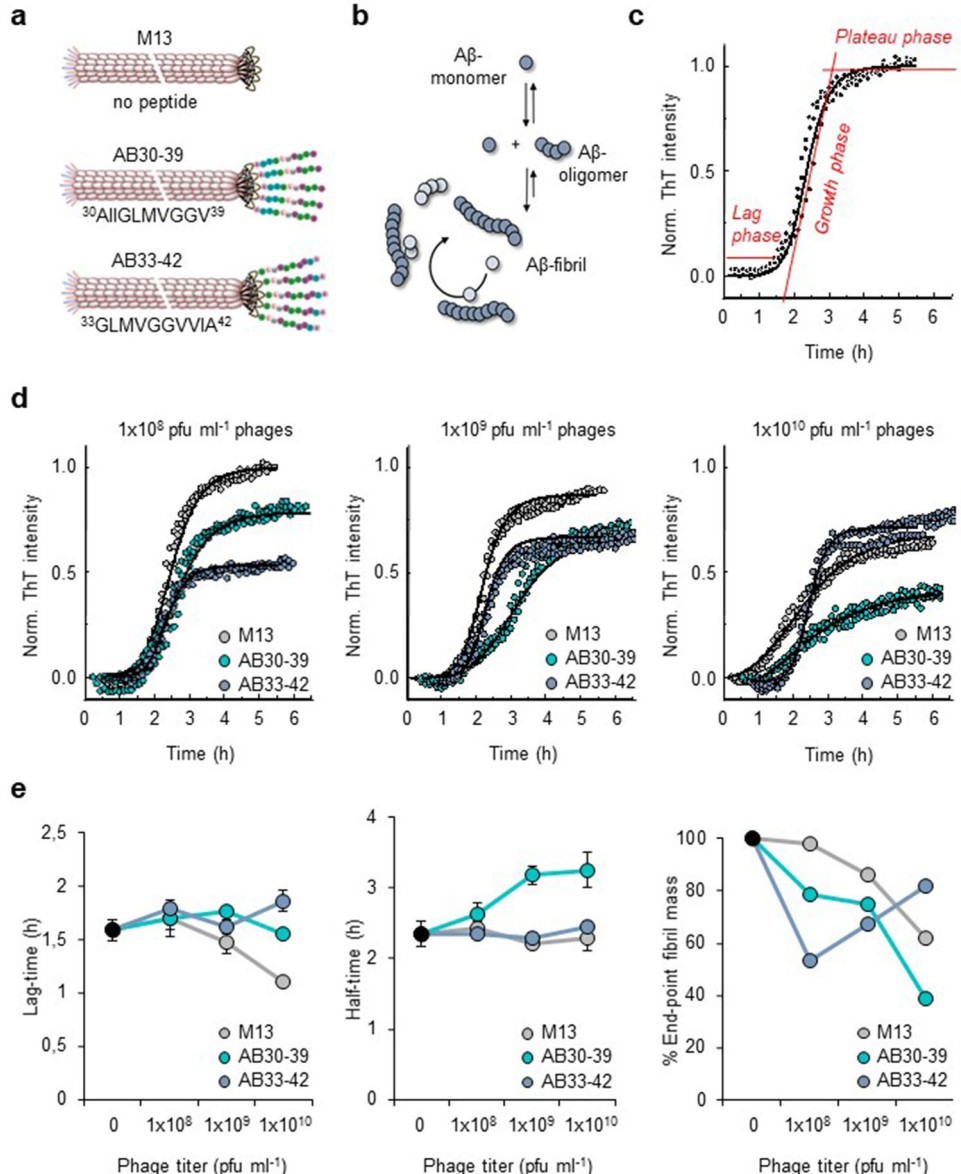

**Fig. 1 Effects of engineered phages on Aβ-aggregation kinetics. a** Schematics depicting M13 and AB-phages. **b** Schematics of Aβ-aggregation process from monomers to oligomers and fibrils. **c** Fibril formation kinetics measured with ThT-fluorescence. **d** Effects of M13 (gray), AB30-39 (green) and AB33-42 (blue) at phage concentration of $1 \times 10^8$ pfu ml$^{-1}$, $1 \times 10^9$ pfu ml$^{-1}$ or $1 \times 10^{10}$ pfu ml$^{-1}$ on fibril formation kinetics measured with ThT fluorescence. **e** Quantification of lag-time, growth-phase half-time, and % end-point plateau levels of fibril formation ($n = 3$ independent experiments). Selective effect of AB30-39 and AB33-42 at $1 \times 10^8$ pfu ml$^{-1}$ on end-point plateau levels suggests AB30-39 and AB33-42 selectively interact with Aβ-oligomers. Data are mean ± SD.

degradation, or clearance of Aβ peptides or oligomers occurs differently at CA1 cell bodies than at dendrites at an early age[19–21].

Staining with AB30-39 and AB33-42 is largely absent in brain tissue from wild-types, although low-level staining was noted in aged animals, in agreement with aged wild-type mice having ~10-fold fewer Aβ-oligomers compared with APP/PS1-mice[22]. These data indicate that AB30-39 and AB33-42 can be used to detect small Aβ-aggregates in brain slices of mice. Notably, no staining was observed when brain samples were either thermally denatured (for antigen retrieval) or fixated with paraformaldehyde for extended time (Supplementary Fig. 1), suggesting that these treatments perturbed the conformation of epitopes at Aβ-oligomers recognized by AB30-39 or AB33-42.

**Accumulation of Aβ-oligomers in brain tissue from APP/PS1-transgenic mice.** Since we observed that Aβ-aggregates were detectable in mice as early as 3-4 months of age, we next examined how the presence Aβ-oligomers develops from an early age. Brain slices were prepared from wild-type and littermate APP/PS1 mice aged 1.5, 3, and 6 months, and the slices were exposed to $1 \times 10^8$ pfu ml$^{-1}$ of phages and subsequently stained with anti-bacteriophage mAbs. AB30-39 and AB33-42, but not M13 control phages, were able to detect Aβ-aggregates in CA1 of 1.5-month-old APP/PS1 mice (Fig. 3a). The level of staining increased with age, and was lower in wild-type littermates (Fig. 3b). Also, in other brain areas that are known to develop Aβ-plaques in APP/PS1 mice at later ages, such as the dentate gyrus, entorhinal cortex, and somatosensory cortex[23], these phages detected the presence of punctate Aβ-aggregates (Supplementary Figs. 2–4).

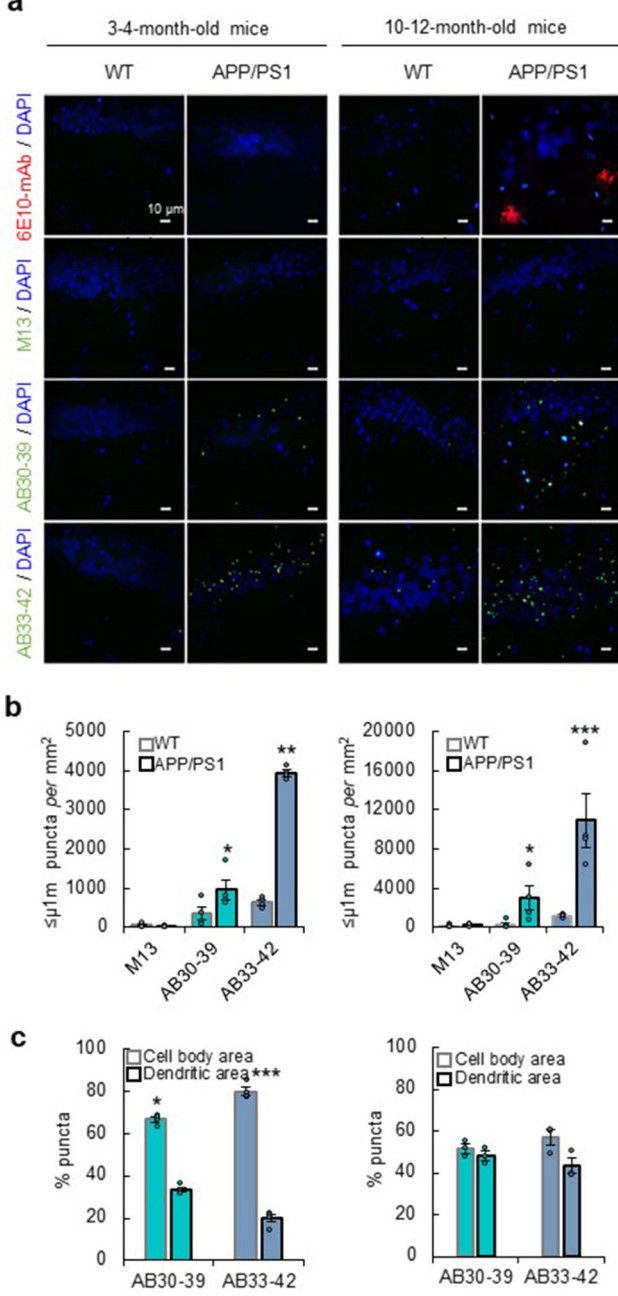

**Fig. 2 Detection of Aβ-oligomers in brain tissue from APP/PS1-transgenic mice. a** Representative immunostaining examples of hippocampal tissue from wild-type and APP/PS1-mice with Aβ-mAb (6E10, red), phages (green), or DAPI (blue). **b** Quantification of ≤1 μm puncta density in CA1 after exposure to $1 \times 10^8$ pfu ml⁻¹ M13 (gray), AB30-39 (green) and AB33-42 (blue) of 3–4-month-old mice (left, $n = 4$) and 10-12-month-old mice (right, $n = 4$). **c** Percentage of puncta density divided between cell body versus dendritic area for 3–4-month-old mice (left, $n = 4$) and 10-12-month-old mice (right, $n = 3$). Data are represented as a mean ± SEM. Statistics: Students $t$-test, $*p \leq 0.05$, $**p \leq 0.01$ and $***p \leq 0.001$.

To assess whether the phages detect Aβ-fibrils, we stained brain slices with the anti-amyloid fibrils OC antibody. Unlike AB30-39 and AB33-42 phages, this OC antibody detected plaque-sized aggregates in 6-month-old mice and somewhat smaller aggregates in 3-month-old mice but did not show any immunostaining in slices of 1.5-month-old mice (Fig. 3a and Supplementary Figs. 2

and 3). The OC antibody did not recognize 1 μm-sized puncta, suggesting AB30-39 and AB33-42 phages do not recognize fibrillar forms of Aβ.

**Engineered bacteriophages detect Aβ-aggregates in brain tissue from AD patients.** To assess whether phages can detect Aβ-aggregates in human brain tissue, samples from hippocampi of three AD-patients and three age-matched non-demented individuals were selected (Supplementary Table 1). Aβ-mAbs showed staining of amyloid-plaques in hippocampi of AD-patients, but minimally in hippocampi from age-matched controls (Fig. 4a). The OC antibody also showed staining of amyloid-plaques but showed limited punctate staining (Fig. 4a). Both AB30-39 and AB33-42 revealed distinctive punctate staining in hippocampi of AD-patients (Fig. 4a, b). Besides ~1 μm-sized puncta, AB-phages also detected larger aggregates (up to ~10 μm) in human tissue, but not plaque-sized inclusions (Fig. 4a, b). Non-demented individuals also showed punctate staining in hippocampi, albeit at lower levels compared with AD-patients (Fig. 4b), corroborating that the presence of Aβ-oligomers in hippocampi of non-demented individuals is not uncommon[22]. These data indicate that AB30-39 and AB33-42 phages can be used to detect small Aβ-aggregates in human *post-mortem* brain tissue.

Our results show that AB30-39 and particularly AB33-42 phages perform well as immunohistochemistry tools to study Aβ-aggregates in *post-mortem* brain tissue of APP/PS1 mice and of AD-patients. They open the possibility of quantifying the levels and identifying the locations of small Aβ-oligomers up to the size of protofibrils in fundamental and clinical AD pathology research. Earlier studies used antibodies or functionalized peptides as tools to detect Aβ-aggregates in brain tissue[22,24,25]. Advantages of phages are that they are convenient to use and can be produced relatively easily and at low cost in large quantities. We caution that, for AB-phages to efficiently detect Aβ-aggregates, brain samples need to be fixated such that the conformation of Aβ-aggregates in the tissue is largely preserved.

Future research may reveal whether phage technology also has potential as a diagnostic or therapeutic method to selectively target Aβ-oligomers and fibrils in vivo as an alternative to currently used antibody-based approaches. Bacteriophages are generally well tolerated by the immune system[26], have the capacity to cross the blood–brain-barrier (BBB)[27], and these characteristics may be further optimized by phage display technology. For future in vivo studies, AB30-39 seems the preferred choice above AB33-42, since our in vitro data show that at high concentrations the former prevents, whereas the latter promotes Aβ-aggregation.

In conclusion, we present a biotechnological tool that detects the presence of Aβ-oligomers in *post-mortem* brain tissue. This phage-based tool may prove valuable for AD-research to study, for instance, correlations between Aβ-oligomer levels and cognitive decline during disease progression.

## Methods

**Reagents.** All reagents were of the highest grade commercially available. Thioflavin T (ThT) was obtained from Sigma. A Chelex resin (Bio-Rad) was used to remove contaminant trace metals from all solutions. Human Aβ42 expression plasmid, as in[28] was a kind gift from J. Presto (Karolinska Institutet, Sweden), to C. Gomes. Recombinant Aβ42 was expressed in *E. coli* and purified as previously described[28]. To obtain the monomeric form, 4 mg of Aβ42 was dissolved in 7 M guanidine hydrochloride and eluted in a Superdex S75 (GE Healthcare) with 50 mM HEPES pH 7.4 and used immediately. Amyloid fibrils of Aβ42 were prepared by incubation of 5 μM Aβ42 at 37 °C for 24 h under quiescent

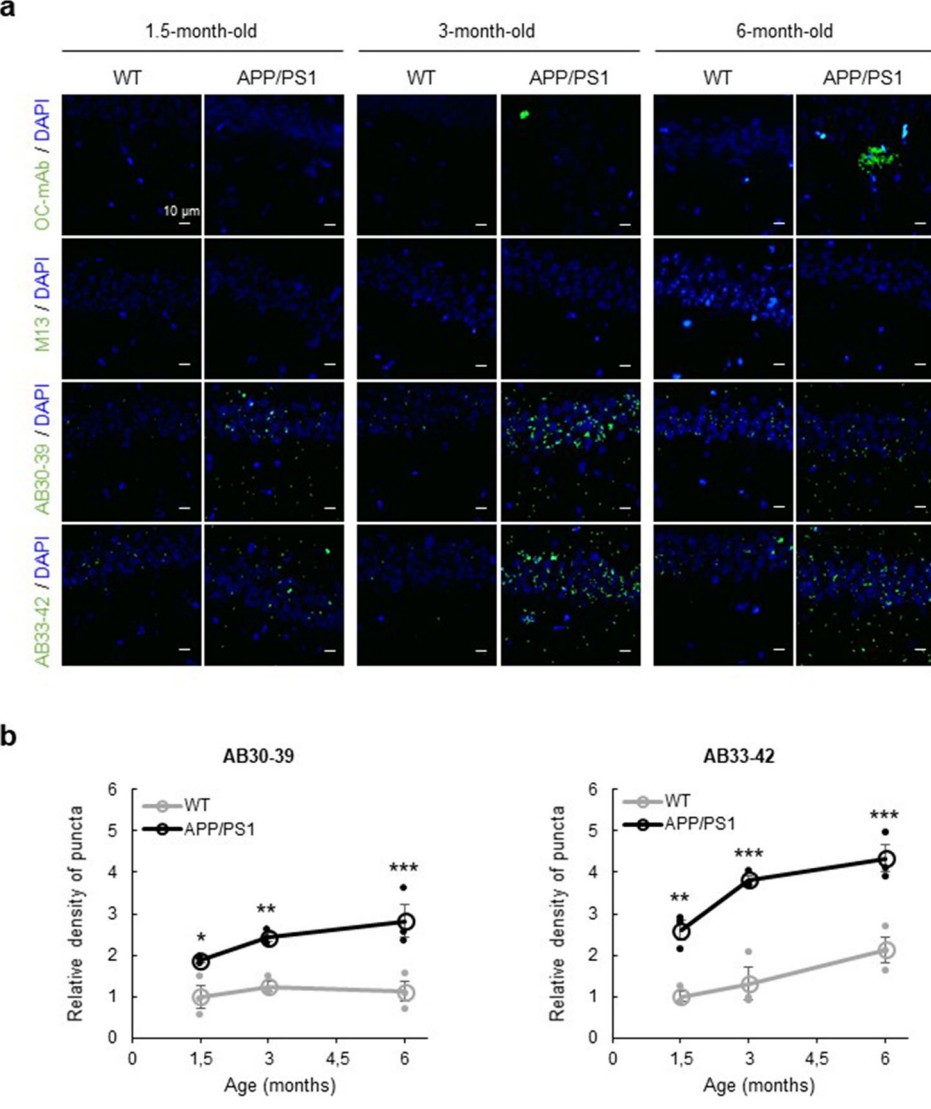

**Fig. 3 Accumulation of Aβ-oligomers in CA1 hippocampus of APP/PS1-transgenic mice. a** Representative immunostaining examples of CA1 hippocampal tissue from 1.5-, 3-, and 6-month-old wild-type (WT) and APP/PS1-mice with OC-mAb or phages (green) and DAPI (blue). **b** Density of ≤1 μm puncta (normalized to 1.5-month-old WT) in CA1, from WT (gray, $n = 3$ animals) and APP/PS1 mice (black, $n = 3$ animals) after exposure to AB30-39 (left) or AB33-42 (right). Data are mean ± SEM. Statistics: Students $t$-test with Šidák correction for multiple comparisons, *$p < 0.05$, **$p < 0.01$ and ***$p < 0.001$.

conditions. Low-binding tubes (Axygen Scientific, Corning) were used in all Aβ42 manipulations.

**Genetic manipulation of M13 phage**. Two amyloidogenic peptide residues of 10 amino acids (Aβ-based) were primer designed to be cloned into the genome of the M13KE phage. Aβ30-39 (AIIGLMVGGV) and Aβ33-42 (GLMVGGVVIA) peptide motifs from the Aβ42 peptide, were genetically fused to the N-terminus of the gene 3, leading to peptide expression on the coat protein III (Fig. 1a). The approach used to clone both peptides in the genome of M13 was based on the phagemid cloning system[29]. Basic components of a phagemid include the replication origin of a plasmid, the selective marker (usually an antibiotic resistance marker), the intergenic region (IG region, usually contains the packing sequence and replication origin of minus and plus strands), a gene of a phage coat protein, restriction enzymes recognition sites, a promoter and a DNA segment encoding a signal peptide[29]. For the phagemid construction, the commercial plasmid pETDuet-1 (Novagen, Darmstadt, Germany) was kindly provided by L. Rodrigues (Centre of Biological Engineering,

Portugal), to I. Martins. This plasmid contains the ampicillin-resistant gene; the T7 promoter (allows gene transcription that affects the expression level of fusion genes); the signal peptide pelB (facilitates the translocation through the bacterial membrane of phage proteins and their assembly in phage particles) and the gene 3 of the M13 phage (codes for the phage coat protein III). The Aβ peptide sequences were cloned immediately before gene 3 of the M13 on the MCS-1 of the pETDuet-1 plasmid between SalI–SacI restriction sites. This recombinant plasmid was transformed into *E. coli* competent cells, and positive clones were confirmed by PCR and sequencing.

Because phagemids can be converted to phage particles with the same morphology by co-infection with helper phages, the kanamycin-resistant M13KO7 helper phage (N0315S, NewEngland BioLabs® Inc), a derivative of the M13 phage containing the kanamycin resistance gene, was used. An infection protocol to merge gene 3 of the plasmid with gene 3 of the helper phage, allowing the display of the Aβ sequence in the phage coat protein III, was performed following the protocol from New England Biolabs. Briefly, the cells containing the phagemid were grown in

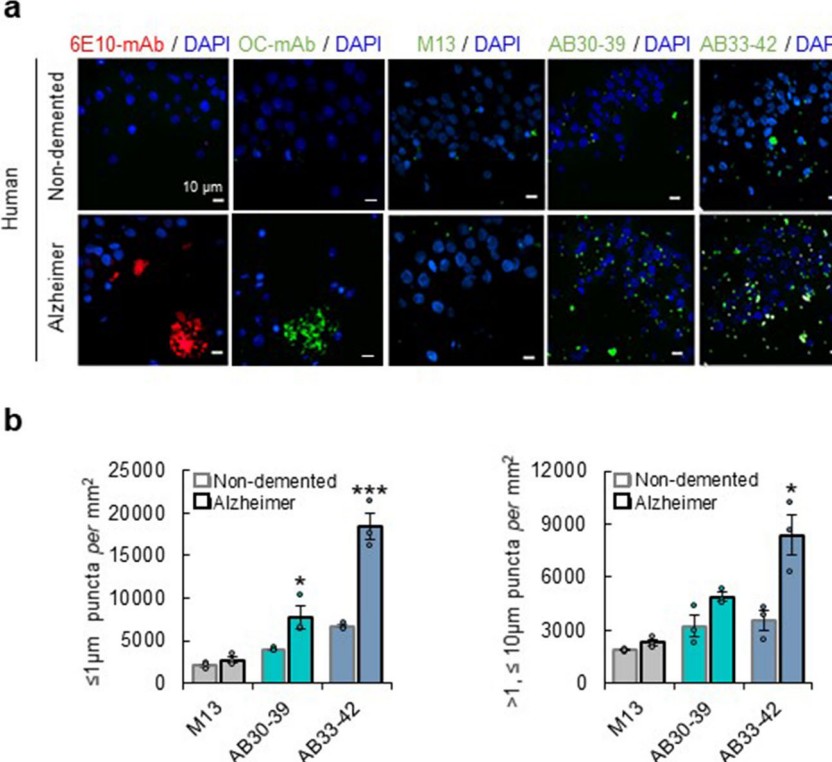

**Fig. 4 Detection of Aβ-oligomers in *post-mortem* brain tissue from AD-patients. a** Representative immunostaining examples of hippocampal tissue from AD-patients and age-match controls with Aβ-mAb (6E10 red and OC green), phages-mAb (green), or DAPI (blue). **b** Quantification of ≤1 μm puncta density in CA1 after exposure to $1 \times 10^8$ pfu ml$^{-1}$ M13 (gray), AB30-39 (green) and AB33-42 (blue) of human hippocampal tissue. Quantification of ≤1 μm puncta (top) and of >1 μm < 10 μm puncta (bottom) in human samples ($n = 3$). Data are mean ± SEM. Statistics: Students *t*-test, *$p < 0.05$, **$p < 0.01$ and ***$p < 0.001$.

LB medium with ampicillin at a final concentration of 20 mg/ml and infected with 50 μl of the M13KO7 helper phage ($1 \times 10^8$ pfu ml$^{-1}$) at 37 °C, 250 rpm for 90 min. Then, kanamycin was added to a final concentration of 70 μg/ml, and the solution was incubated overnight at 37 °C and 250 rpm. To separate bacteria cells from the phages a centrifugation at 6000 rpm for 10 min was performed and the supernatant was transferred to a new tube. Phage was precipitated by the addition of PEG/NaCl solution followed by incubation at 4 °C for 2 h and resuspended in Tris-buffered saline (TBS). Phage genomic ssDNA was isolated using an equal volume of phenol–chloroform–isoamyl alcohol (25:24:1, v/v), purified with an equal volume of chloroform, precipitated with 100% ethanol, and resuspended in Tris-EDTA (TE). To check the Aβ sequences, the DNA was sequenced using a primer that amplified the region of interest of the gene 3. Phage titration was performed following the double agar overlay technique. Briefly, 10 μl of serially diluted phage, 200 μl of host bacteria culture, and 3 ml of soft agar were mixed and poured onto an LB plate with ampicillin. After overnight incubation at 37 °C, the plaque-forming units (pfus) were determined.

**Aβ42 aggregation kinetics**. Aggregation kinetics were performed recording the ThT fluorescence intensity as a function of time in a plate reader (Fluostar Optima, BMG Labtech) with a 440 nm excitation filter and a 480 nm emission filter, as *per* original reports[10,11]. The fluorescence was recorded using bottom optics in half-area 96-well polyethylene glycol-coated black polystyrene plates with a clear bottom (3881, Corning). Aβ42 monomer was isolated by gel filtration (Tricorn Superdex75 column, GE Healthcare) in 50 mM HEPES, pH 7.4. Highly homogeneous preparations of monomeric Aβ42 were used within each

experiment to ensure highly reproducible results in aggregation kinetics[10,11,30]. 10 μM of ThT was added to each condition. AB phages and M13 were added at the start of the reaction at different titers ($1 \times 10^8$, $1 \times 10^9$, and $1 \times 10^{10}$ pfu ml$^{-1}$). Assays were performed in triplicates at 37 °C, without agitation with fluorescence read every 400 s using well-established and highly reproducible procedures[31–33]. Data from three independent experiments were analyzed using Amylofit[30] according to the well-established secondary nucleation mechanism[10]. From this analysis, one can obtain the reaction half-time values. Plots were made in Origin. Fibril mass at endpoints was averaged from fluorescence recordings at the post-transition plateau stage.

**Immunofluorescence assays**. Brains of age-matched APPswe/PS1dE9 transgenic and wild-type mice were isolated in ice-cold phosphate-buffered saline (PBS) pH 7.45, deep-frozen with liquid nitrogen and stored at −20 °C. Brain tissue was sectioned using a cryostat (CM3050S, Leica) with chamber temperature −15 °C and object temperature −13 °C. The cutting angle was set to 5 degrees and the size of the sections to 10 μm. The tissue was trimmed to reach the hippocampal region. The sections were mounted on SuperFrost Plus glass (11950657, Thermo Scientific) and stored at −20 °C. Animal experiments were conducted under Dutch national law and in compliance with the European Union Directive 2010/63/EU. The study design was evaluated and approved by the animal welfare committee of the University of Amsterdam.

*Post-mortem* human hippocampal 7 μm sections from AD patients and controls were kindly provided by the Department of Neuropathology from Amsterdam UMC (Supplementary Table 1).

Cryopreserved *post-mortem* slides were fixated for 10 min with 4% paraformaldehyde (PFA) in PBS. Slides were incubated for 1 hour at room temperature with blocking solution of 2.5% bovine serum albumin (BSA) (Sigma-Aldrich, 10735086001) in TBS plus 0.1% of Tween-20 (TBST). Next, slides were incubated with phages at a concentration of $1 \times 10^8$ pfu ml$^{-1}$ in TBST overnight at 4 °C in a humidified chamber. Slides were washed in TBST five times for 10 minutes and subsequently incubated overnight at 4 °C in a humidified chamber with 1:1000 diluted rabbit anti-fd phage antibody (B7786, Sigma). The slides were then washed and incubated for 2 h at room temperature with 1:200 diluted FITC-labeled goat anti-rabbit IgG antibody (F9887, Sigma). For 6E10 and OC antibody staining, antigen retrieval was performed by heat-mediated treatment with sodium citrate buffer 10 mM pH 6.0 (S1894-500G, Sigma-Aldrich), where a boiling state was set for 20 min followed by a cooling down to room temperature for 20 minutes. Slides were incubated with 1:5000 diluted mouse anti-β-amyloid 6E10 antibody (80300, BioLegend) followed by 1:700 diluted donkey anti-mouse IgG (H + L) labeled with Alexa Fluor 594 (A-21203, Thermo Scientific), or with 1:1000 diluted anti-amyloid fibrils OC antibody (AB2286, Merck) followed by 1:200 diluted FITC-labeled goat anti-rabbit IgG antibody. All antibodies were diluted in TBST with 1% BSA. Slides were covered with Vectashield mounting medium with DAPI (VectorLabs). Images were acquired with a fluorescence microscope (Nikon Eclipse E400) and analyzed using ImageJ[34]. The quantification was performed in ×40 amplification images using a color threshold set to identify the fluorescein signal (phages) and all speckles within a defined size threshold between 10-150 pixels were counted for mouse and human samples.

**Statistics and reproducibility**. Sample sizes were based on power analyses of preliminary data. Data were evaluated by the Student's *t*-test with Šidák correction for multiple comparisons, $p < 0.05$ was considered statistically significant for all analyses. Figure 1 presents ThT-monitored Aβ42 aggregation kinetic curves obtained from fitting data gathered from three independent experiments to a secondary nucleation mechanism. Because the fits of these curves are statistically robust, no other type of statistical analysis between estimated kinetics values for different conditions is used.

**Reporting summary**. Further information on research design is available in the Nature Portfolio Reporting Summary linked to this article.

## Data availability
The source data behind the graphs can be found in Supplementary Data.

## Material availability
The microscopy images are available from the corresponding authors upon reasonable request. The generated phagemids will be available from the corresponding authors upon request through a Material Transfer Agreement (MTA).

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

## Acknowledgements

We acknowledge Mariana Cardoso Resende, Tessa Lodder, and Shakira van der Panne for their contribution to the optimization of experiments. This study was supported by the Portuguese Foundation for Science and Technology (FCT) under the scope of the strategic funding of UIDB/04469/2020 unit, center grant UID/MULTI/04046/2020 (to BioISI), Ph.D. fellowship SFRH/BD/101171/2014 (to J.S.C.), by Brain Foundation Netherlands (Hersenstichting; to H.W.K.), and by Alzheimer's Society in the Netherlands (Alzheimer Nederland; to H.W.K.).

## Author contributions

I.M.M., L.D.K., and H.W.K. conceived the study. I.M.M., C.M.G., J.A., and H.W.K. designed the experiments. I.M.M., A.L., W.d.G., J.S.C., and N.B. performed the experiments. I.M.M., A.L., C.M.G. and H.W.K. analyzed data. E.A. provided human brain samples. I.M.M., C.M.G., and H.W.K wrote the manuscript. All authors reviewed the manuscript.

## Competing interests

The authors declare no competing interests.
