## [Peer Review File · Communications Biology]

Reviewers' comments:

Reviewer #1 (Remarks to the Author):

This is an interesting study in which the author tries to demonstrate a new tool to identify oligomeric aggregates in brain tissue from Tg mice and human AD cases. In this study the author have generated two phages that contain Ab-derived peptides on their surface and they are capable to detect the formation of small Ab-oligomers aggregates. The current study seems highly appealing, however, several comments should be addressed to clarify the nature of these small aggregates recognized by these phages.

1. The author must consider to perform some double immunohistochemistry in Tg mice and human cases with the OC antibody and AB30-39 and AB33-42. They should clarify whether or not these small aggregates can be also recognized by the OC antibody which is a specific antibody for fibrillar oligomeric forms of Ab.
2. Why 6E10 antibody is not able to recognize any plaque in APP/PS1 mice without antigen retrieval. Many different studies have shown that it is not necessary antigen retrieval to identify plaques with 6E10 in many different Tg-AD mice.
3. The author should consider to show the brain-pattern of expression and accumulation of these oligomeric aggregates. It will be interesting to investigate in which brain region it appears first and propagates.

Reviewer #2 (Remarks to the Author):

In the present article, the authors have implemented the A β peptide fragments tagged on the coat of M13 bacteriophage to detect A β oligomers which occur in the brain of AD animals prior to the formation of large A β plaques. Eventually these A β peptide tagged M13 phages were also shown to bind the A β oligomers present in the human postmortem brains. Overall, the approach of using M13 phage-based technology as an alternative to oligomeric A β specific antibodies for experimental purpose seems novel and might be easily implemented in the laboratories. However, there are certain concerns that needs attention and should be addressed for improving the quality of this article.

1. The authors performed in vitro experiments (ThioT assay) to show inhibition of A β aggregation by the peptide fragments 30-39 and 33-42, whereas as the control they have used only M13 phage. However, to prove the specificity of these two specific peptides in binding A β oligomers, the authors should have used a control peptide of similar length having no propensity to bind A β oligomers.
2. Also, in Fig 1 there is no statistical analyses shown to provide the significance between samples. Opposite effects on fibrillization by lower and higher doses of M13 AB30-39 is confusing and needs proper clarification.
3. How did the authors distinguish between the A β oligomers present in the cell body area and dendritic area as no specific markers have been used in the immunofluorescence.
4. Fig S1 demonstrates that without antigen retrieval there is no signal of 6E10 immunofluorescence in the APP/PS1 sections, which is not aligning with the existing reports.
5. For in vivo studies, have the authors conducted any power analysis to determine the number of animals in each group, such as n=4 in Fig 2?
6. The authors stated that APP/PS1 mice starts experiencing synaptic deficits and memory impairments at 3-4 months of age (line 113-114) is incorrect. The references provided by them demonstrated LTP impairments and loss of cognitive skills only at 6 months of age and it increases along with aging. Therefore, this statement needs correction.
7. The authors also stated the presence of glial cells in the vicinity of dendritic areas of the CA1 region

of hippocampus. However, there is no connection found between the present report and the glial clearance of A β . Moreover, the authors did not show any comprehensive immunostaining of dendritic markers to locate the A β oligomers and their close association with the glial cells. It is advised to remove this unnecessary information from the text.

Response to Reviewers Comments

A point-by-point response to the reviewers' comments is given below in blue and the changes made to the original manuscript reflecting their suggestions are highlighted in yellow.

Reviewer #1 (Remarks to the Author):

"This is an interesting study in which the author tries to demonstrate a new tool to identify oligomeric aggregates in brain tissue from Tg mice and human AD cases. In this study the author have generated two phages that contain Ab-derived peptides on their surface and they are capable to detect the formation of small Ab-oligomers aggregates. The current study seems highly appealing, however, several comments should be addressed to clarify the nature of these small aggregates recognized by theses phages."

1. The author must consider to perform some double immunohistochemistry in Tg mice and human cases with the OC antibody and AB30-39 and AB33-42. They should clarify whether or not theses small aggregates can be also recognized by the OC antibody which is a specific antibody for fibrillar oligomeric forms of Ab.

We thank the reviewer for this suggestion. We have performed immunofluorescence assays with the OC antibody on both mouse and human samples to complement our previous results. We found that OC antibodies detect plaque-sized aggregates in 6-month old mice and somewhat smaller aggregates in 3-month old mice, but did not show any immunostaining in slices from 1.5-month old mice (new Figure 3A in revised manuscript). Importantly, the OC antibody did not recognize 1 μ m-sized puncta, suggesting AB30-39 and AB33-42 phages do not recognize fibrillar forms of A β . Also in human samples OC-staining predominantly stains plaques, which is clearly distinct from phage staining (Figure 4A of revised manuscript).

We note firstly that OC-antibodies require antigen retrieval while for phages antigen retrieval needs to be avoided, and secondly that both OC antibodies and anti-phage antibodies use the same secondary (i.e. anti-rabbit) antibody. As a consequence, double-staining with OC antibodies and phages was not possible. We instead have alternated staining with phages and OC antibody for successive brain slices, and these results clearly indicate OC-mAb staining and phage staining do not overlap.

The added text is highlighted on pages 5, 6 and 7 of the revised manuscript.

2. Why 6E10 antibody is not able to recognize any plaque in APP/PS1 mice without antigen retrieval. Many different studies have shown that it is not necessary antigen retrieval to identify plaques with 6E10 in many different Tg-AD mice.

Thank you for the pertinent comment. Previous studies used the 6E10 antibody on brain slices that were fixated for extended time periods (i.e. hours), and also in our hands 6E10 mAbs clearly stain plaques without the need for antigen retrieval on slices that are fixated for 24 hrs. However, phage staining requires short fixation (10 min), and in those conditions 6E10 only recognizes plaques provided that brain samples were thermally denatured for antigen retrieval (Figure S1 of the revised manuscript).

3. The author should consider to show the brain-pattern of expression and accumulation of these oligomeric aggregates. It will be interesting to investigate in which brain region it appears first and propagates.

We performed immunofluorescence assays in different brain regions (CA1, dentate gyrus, entorhinal region, and neocortex) of APP/PS1 and WT littermate mice at three different ages (1.5-, 3- and 6-months old). We observed that already at the youngest age of 1.5 months the oligomeric aggregates could be detected, albeit at a lower level than at older ages, indicating gradual accumulation (new Figure 3 in revised manuscript). In addition, the patterns of staining of these aggregates are similar across different brain regions of APP/PS1-transgenic mice (new Figures S2, S3 and S43 in revised manuscript).

The added text is highlighted at pages 6 and 7 of the revised manuscript.

Reviewer #2 (Remarks to the Author):

“In the present article, the authors have implemented the A β peptide fragments tagged on the coat of M13 bacteriophage to detect A β oligomers which occur in the brain of AD animals prior to the formation of large A β plaques. Eventually these A β peptide tagged M13 phages were also shown to bind the A β oligomers present in the human postmortem brains. Overall, the approach of using M13 phage-based technology as an alternative to oligomeric A β specific antibodies for experimental purpose seems novel and might be easily implemented in the laboratories. However, there are certain concerns that needs attention and should be addressed for improving the quality of this article.”

1. The authors performed in vitro experiments (ThioT assay) to show inhibition of A β aggregation by the peptide fragments 30-39 and 33-42, whereas as the control they have used only M13 phage. However, to prove the specificity of these two specific peptides in binding A β oligomers, the authors should have used a control peptide of similar length having no propensity to bind A β oligomers.

We appreciate the opportunity to explain why we indeed have not used a control peptide of similar length. Our study is inspired and based on a previous study from the Tessier lab that showed that A β peptides 30-39 and 33-42 selectively interact with A β -oligomers and fibrils, whereas other A β -derived 10-mer peptides do not (ref 7: Perchiacca et al, PNAS 2011). Our data show that when these two different peptides are grafted on the surface of M13 phages, they have distinct and opposite effects on A β fibrillization: while AB30-39 leads to lower fibril mass, AB33-42 promotes a higher fibril mass (Figure 1d,e), further indicating that not any 10 amino acid peptide interferes with A β fibrillization in the same way.

We explained these distinct actions of the two phages more clearly in highlighted text at page 4 of the revised manuscript.

2. Also, in Fig 1 there is no statistical analyses shown to provide the significance between samples. Opposite effects on fibrillization by lower and higher doses of M13 AB30-39 is confusing and needs proper clarification.

Figure 1d presents ThT-monitored A β 42 aggregation kinetic curves (solid lines) obtained from fitting data gathered from three independent experiments (dots) to a secondary nucleation mechanism, as per the best standards in the literature and the usual statistical analysis (cited refs 29-31 and also for e.g. PMID: PMC9732875 or 35580187). Because the fits of these curves are statistically robust, neither we nor other reference laboratories that carry out these type of experiments use any other type of statistical/significance analysis between estimated kinetics values for different conditions.

From this analysis one extracts the time at which the fibrillization reaction proceeds at half completion (half-time, middle panel in Figure 1e), while the relative variation of the ThT intensity at the end-point indicates the effect on fibril mass (right panel in Figure 1e). In the presence of AB30-39, data show a slight increase in the reaction half time with increasing titers (denoting an increasing delay in aggregation) and a corresponding decrease in the amount of formed fibrils. Therefore, no opposite effects on fibrillization are observed at lower and higher

doses of M13 AB30-39: What the middle and right panels in Figure 1e depict is the increase in half times (middle panel) and the decrease in fibril mass (right panel) at increasing titers. In the presence of AB33-42, at low concentrations we observed a substantial decrease in fibril mass, while at higher concentrations the fibril mass is decreased less. The explanation for this observation is that at low concentration AB33-42 only interferes with the formation of A β -oligomers but not fibrils.

To clarify this, we have made changes in text (high-lighted) in the Results and Discussion section at pages 4 and 5, and in the Methods section at pages 10/11 of the revised manuscript.

3. How did the authors distinguish between the A β oligomers present in the cell body area and dendritic area as no specific markers have been used in the immunofluorescence.

We used DAPI staining (in blue) to visualize the nuclei of pyramidal neurons in the cell body area of the CA1 region. Based on this marker, we were able to distinguish between cell body area and dendritic area with CA1. This process was performed uniformly across all samples.

4. Fig S1 demonstrates that without antigen retrieval there is no signal of 6E10 immunofluorescence in the APP/PS1 sections, which is not aligning with the existing reports.

Thank you for pointing this out, because this point was also raised by reviewer 1, we here copy the same answer: Previous studies used the 6E10 antibody on brain slices that were fixated with extended fixation period of time, and also in our hands 6E10 mAbs clearly stain plaques without the need for antigen retrieval on slices that are fixated for 24 hrs. However, phage staining requires short fixation, and in those conditions 6E10 only recognizes plaques provided that brain samples were thermally denatured for antigen retrieval (Figure S1 of the revised manuscript).

5. For in vivo studies, have the authors conducted any power analysis to determine the number of animals in each group, such as n=4 in Fig 2?

We determined the number of animals (n=4) for experiments shown in Figure 2 using a power analysis based on our preliminary data in which we tested the optimal conditions for staining. For experiments shown in the new Figure 3 of the revised manuscript we used n=3 based on data shown in Figure 2.

6. The authors stated that APP/PS1 mice starts experiencing synaptic deficits and memory impairments at 3-4 months of age (line 113-114) is incorrect. The references provided by them demonstrated LTP impairments and loss of cognitive skills only at 6 months of age and it increases along with aging. Therefore, this statement needs correction.

Reference 17 (Trinchese et al) shows in Figure 2 that spatial working memory is impaired for APP/PS1 mice at 3-4 months of age, but not at 2 months of age.

Reference 18 (Reinders et al) is a paper from our group. Indeed, main figure 6 shows memory deficits (contextual fear conditioning) for 6 and 12 month old mice, but Supplemental Figure 6 shows that also at 3 month old APP/PS1 mice display a memory deficit.

(<https://www.pnas.org/doi/full/10.1073/pnas.1614249113#supplementary-materials>)

We prefer to include this citation, since these data are obtained from the same colony of mice as the data in this manuscript, despite this being supporting information.

We added a new reference, reference 19 (Vegh et al), which shows in Figures 2 and 4 that APP/PS1 mice have impaired hippocampus-dependent reference memory (morris water maze) when 3 months old.

7. The authors also stated the presence of glial cells in the vicinity of dendritic areas of the CA1 region of hippocampus. However, there is no connection found between the present report and the glial clearance of A β . Moreover, the authors did not show any comprehensive immunostaining of dendritic markers to locate the A β oligomers and their close association with the glial cells. It is advised to remove this unnecessary information from the text.

We completely agree and have removed the indicated sentence. The point we wanted to make is that we don't know whether the difference is due to changes in the production, degradation or clearance of A β . In the highlighted text at page 6 of the revised manuscript, we now state: "Possibly the production, degradation or clearance of A β peptides or oligomers occurs differently at CA1 cell bodies than at dendrites at an early age".

REVIEWERS' COMMENTS:

Reviewer #1 (Remarks to the Author):

The authors have addressed my comments properly. Therefore, the manuscript should be considered for publication in this journal.

Reviewer #2 (Remarks to the Author):

The authors have addressed my concerns and justified it with references.